

# Identifying successful combinations by fertility index in old garden roses and hybrid tea roses crosses

Tuğba Kılıç

Horticulture Department, Yozgat Bozok University, Yozgat, Turkey

## ABSTRACT

The success of rose breeding programs is low due to poor seed sets and germination rates. Determining fertile parents and cross combinations that show high compatibility could increase the effectiveness of breeding programs. In this study, three rose varieties belonging to *Rosa × hybrida* (Jumilia, First Red and Magnum), and two old garden rose species (Black Rose and Cabbage Rose) with known ploidy levels were reciprocally crossbred under controlled conditions to determine the successful crosses by checking fertility. The pollen germination rate (PG), crossability rate (CR), seed number per fruit (SNpF), seed production efficiency (SPE), seed germination rate (SGR), fruit weight (FW), seed weight (SW) and stigma number (SiN), *etc.* were recorded. Comprehensive fertility index value was calculated. Principal component analysis (PCA), correlation matrix, and hierarchical heat map were used to evaluate the data. The findings showed that old garden roses had more viable pollen than hybrid tea roses. The crossing success improved as pollen fertility increased. Also, female parent fertility improved crossing success just as much as pollen fertility. Although the pollen fertility and stigma numbers were low, some combinations had higher CR and SPE. The maximum SPE (from 8.67% to 19.46%) was determined in combinations where Black Rose was the female parent despite the lower stigma number and low pollen fertility. The highest CR was recorded in Black Rose × First Red (94.36%). All combinations in which Black Rose was used as the female parent had a more stable CR. The SNpF of combinations where hybrid rose varieties were female parents and old garden roses were pollen parents was higher than other combinations where hybrid rose varieties were both female and pollen parents. The SPE in intraspecific crosses was lower than that obtained from interspecific crosses. Moreover, the SGR decreased in combinations that produced heavier seeds. The results suggested that SPE is a more accurate parameter than SNpF in demonstrating combination success in breeding programs. Black Rose × First Red, Black Rose × Jumilia, Black Rose × Magnum and Black Rose × Cabbage Rose combinations can be used successfully as the PCA and heat map showed. Black Rose showed better performance as both seed and pollen parents according to the comprehensive fertility index. From the correlation matrix, it is understood that the number of stigmas cannot be an important criterion in parent selection. Old garden roses can be used as parents to increase the success of breeding programs. However, it is necessary to reveal how successful they are in transferring desired characteristics such as scent, petal number, and color.

Corresponding author
Tuğba Kılıç, tugba-klc@hotmail.com

# INTRODUCTION

The rose (*Rosa* spp.) is one of the most popular flowering plant species grown worldwide (*Kılıç, 2020*). It is the most widely cultivated ornamental plant for commercial purposes, and cut roses are highly preferred among roses grown for decorative use. Millions of cut roses of different colors, shapes, and types are produced in more than 50 countries every year (*Trademap, 2022*). Moreover, consumer demands are constantly changing, and thousands of new varieties are developed by breeding companies each year (*Doğan, 2022*). It is imperative for companies to develop new rose varieties to maintain their market share (*Uran, 2022*).

Taxonomic diversity in roses offers breeders the opportunity to be successful in the development of new varieties (*de Vries & Dubois, 1988*). The most efficient way to benefit from this diversity is crossbreeding; therefore, this is the most commonly used breeding method for developing new rose varieties (*Liorzou et al., 2016*). However, roses are known for their difficult sexual reproduction, from pollination to seed set and germination (*Perez & Moore, 1985*; *Gudin, 1992*; *Abdolmohammadi et al., 2014*; *Doğan et al., 2020*). The meiotic abnormalities and accumulation of deleterious recessive alleles due to the interspecific origin and the intensive inbreeding performed in the past have reduced success in cross breeding and represent the economic risk of breeding programs. Moreover, self- and cross-incompatibility, which are widely prevalent in varieties of rose with low pollen quality and seed dormancy, reduce the efficiency of breeding programs (*Debener, Janakiram & Mattiesch, 2000*; *Pipino et al., 2013a*). According to rose breeding studies, some cross combinations do not produce fruit or seeds, and even when they do, the seed germination rate is very low, with some seeds not germinating at all (*Farooq et al., 2016*; *Khan et al., 2020*). However, there are some combinations in which the fruit set and seed germination rate can reach 100.0% (*Grossi & Jay, 2002*; *Atram et al., 2015*) and the seed number can reach 100 per fruit (*Turna, 2022*). A similar trend was found for pollen fertility. It has been stated that pollen germination rates show a wide variation from 0% to 99.0% depending on genotype and climactic conditions (*Pipino et al., 2011*; *Giovannini et al., 2017*).

Considering that the development of new varieties in rose breeding programs takes from 4 to 10 years and the chance of developing new varieties is from 0.002% to 0.003% (*Chaanin, 2003*), rose breeders aspire to increase the efficiency of a breeding program by producing more offspring. Therefore, successful parent selection is of great importance. They must know the probability of producing large viable numbers of seeds in the species and varieties, as well as the compatibility of combinations, the desired characteristics, the ability to transfer the desired characteristics according to the breeding goals like scent, petal number and flower stem length to the next generation, pollen viability and germination rates. Knowing pollen parent fertility and the compatibility of cross combinations is needed to improve the breeding program efficiency and the chance of developing new varieties by

increasing the seed set rate and seed germination rate. Scientific studies on the success of crossing old garden roses and their ability to transfer characteristics to the next generation are limited. The lack of sufficient information is a great challenge for both researchers who will conduct breeding studies and amateurs who will just start breeding roses.

There are thousands of rose varieties and hundreds of rose species. With such a large number of rose genotypes to choose from, defining their fertility becomes difficult, and selecting potential varieties with characteristics takes years (*Khan et al., 2021*). Moreover, there is very little scientific research conducted on the success of crosses in roses because it is mainly carried out by commercial companies, and key information is kept as a trade secret. Modern roses have the desired characteristics in terms of marketability, but their more complex genetic backgrounds suggest that their fertility is lower than that of old garden roses. The crosses between old garden and modern roses are thought to create more successful combinations than crosses among modern roses. In addition, cross incompatibility may be more common in modern roses. This study was designed to identify cross combinations among old garden roses and hybrid tea roses that exhibit good performance in terms of viable seed set, aiming to enhance the effectiveness of breeding programs. Additionally, the study determined parental fertility, incompatibility and the relationships among traits indicating fertility. Enhancing the success rate of hybridization among the existing gene pool of *Rosa* species is essential.

## MATERIALS & METHODS

Crossbreeding on roses was conducted during 2020 and 2021 in a polyethylene (roof cover system-180 µm) and polycarbonate (forehead and side coating-8 mm) plastic-covered greenhouse belonging to the Department of Horticulture in the Agriculture Faculty of Ankara University based in the province of Ankara, Turkey (39°57′53.8″N 32°51′50.8″E (*GoogleMaps, 2020*)). Studies on pollen viability and germination rates of rose species/varieties were conducted in the cytology laboratory of the Department of Horticulture at Ankara University at the same time as the crossbreeding.

### Plant material

The plant material consisted of three of the most widely known global varieties of hybrid tea roses (Jumilia (J), First Red (FR) and Magnum (M)) belonging to the *Rosa × hybrida*, and two different old garden rose species known as Black Rose (*Rosa odorata* Louis XIV, BR), and Cabbage Rose (*Rosa centifolia* L., CR). Hybrid tea roses are the most popular group among modern roses widely used today. They are firm favorite in the cut flower sector with their repeating blooms, standard type, large and semi-double or double flowers (usually 25 to 30, but up to 80 in some hybrids) and long flower stems. Old garden roses are generally not preferred as ornamental plants except for outdoor plant use because they do not meet commercial quality criteria. They are of industrial importance due to their intense scent (*Kılıç, 2020*). While modern roses have the desired higher visual quality, old garden roses have a higher pollen quality than modern roses. Some floral characteristics recorded in the study, such as the petal and stigma numbers of the roses, are given in Table 1.

**Table 1  Average petal and stigma number, and nuclear DNA content of rose genotypes used as pollen and female parent.**

| Rose Species/Varieties | Petal Number | Stigma Number | 2C (pg)[*] | Ploidy[*] |
|---|---|---|---|---|
| Jumilia | 45 | 135 | 2.33 | 4x |
| Magnum | 30 | 160 | 2.54 | 4x |
| First Red | 35 | 400 | 2.36 | 4x |
| Black Rose | 25 | 40 | 2.44 | 4x |
| Cabbage Rose | 45 | 175 | 2.43 | 4x |

Notes.

[*]The petal and stigma number in one flower was counted in three repetitions. The ploidy levels were determined by PARTEC (CyFlow Space) brand flow cytometry device, and the ploidy levels were confirmed by the chromosome counting method. The chromosome numbers were determined by counting the mitotic chromosomes of the cells on the preparations prepared using root tip meristem tissues. In order to determine the ploidy levels, the core DNA contents of the plants were determined first; then, the chromosomes of one of the plants with different DNA content were counted and the DNA content was correlated with the chromosome number, that is, the ploidy level. In the analysis, the core DNA content of three plants was determined in each plant material.

It was important that the ploidy levels were the same in the selection of the parents. All species and varieties were tetraploid with $2n = 4 \times = 28$ chromosome numbers, and the DNA content varied between 2.33 pg/2C and 2.54 pg/2C (*Kılıç, 2020*; *Kazaz et al., 2022*, unpublished data) (Table 1).

All of the rose genotypes used as parents were purchased as 1-year-old seedlings from the Şanlıurfa-based company Atılım in 2017 and brought to the greenhouse, which had a double-row bed system 40 cm high and 20 cm wide. After planting the seedlings in horizontal bags ($100 \times 20 \times 12$ cm, 24 liters) containing the cocopeat, they were placed on beds with a width of 80 cm. Each horizontal bag contained six plants. During the vegetation period, the greenhouse temperature was kept at 23–30 °C, and the relative humidity was maintained at 60%–70%. To prevent the plants from being damaged by the high light intensity, a heat-shade curtain providing 55% shade was used. Water and nutrients were given to the plants by a drip irrigation system, and the system was automatically controlled by a fertigation computer. During the vegetation period, the number of irrigations was adjusted based on 30% drainage rate. The amount of water per drip was generally adjusted to be from 80 cc to 100 cc. The nutrient solution given by *Kılıç (2020)* was used for fertilizing the plants and the electrical conductivity (EC) of the solution given to the plants was kept to be from 1.5 to 1.8 mS/cm in the early stages of development and 1.8 to 2.0 mS/cm in the following periods, and the pH was between 5.3 to 5.8 (*Hazar & Baktır, 2013*). The formulation of the nutrient solutions is given in Table 2. Pesticides and biological control agents were used against diseases and pests. Pesticides were applied to the plants until 7 days before and 15 days after the cross. In the biological control, the predator mite (*Phytoseiulus persimilis*) was used against red spiders.

## Crossbreeding

Reciprocal hybridization (cross pollination) was performed between varieties and species from May 15 to June 30. A total of 20 combinations were formed, and 30 crosses were produced for each combination. First, emasculation was performed on the flowers selected

**Table 2  Nutrient solution formulation.**

| Macro elements | Minimum (mmol/l) | Maximum (mmol/l) |
|---|---|---|
| NO$_3$ | 5.0 | 9.0 |
| NH$_4$ | 0.1 | 0.3 |
| H$_2$PO$_4$ | 0.8 | 1.4 |
| SO$_4$ | 1.0 | 1.6 |
| K | 3.5 | 5.0 |
| Ca | 3.0 | 4.5 |
| Mg | 0.8 | 1.2 |
| **Micro elements** | **($\mu$mol/l)** | **($\mu$mol/l)** |
| Fe | 30.0 | 40.0 |
| Mn | 2.0 | 4.0 |
| Zn | 2.0 | 2.0 |
| B | 20.0 | 30.0 |
| Cu | 1.0 | 1.0 |
| Mo | 1.0 | 1.0 |

Notes.

NO$_3$, nitrate; NH$_4$, ammonium; H$_2$PO$_4$, phosphoric acid; SO$_4$, sulfate; K, potassium; Ca, calcium; Mg, magnesium; Fe, iron; Mn, manganese; Zn, zinc; B, boron; Cu, copper; Mo, molybdenum.

as the female parents of all varieties when one-third of the flowers were opened at 08:00 AM. Then, they were covered with a paper bag (*Crespel & Mouchotte, 2003*; *Chimonidou et al., 2007*). Anthers taken during emasculation were used as pollen parents, and pollen grains were expected to be released from the anther after one day in a growth chamber at a temperature of 24 °C and 60% humidity. 24 h after the emasculation of flowers and collection of anthers, the pollen was dispersed and rubbed onto the stigma with a brush (*Jacob & Ferrero, 2003*; *Spethmann & Feuerhahn, 2003*) and again covered with a paper bag for four days. Labels with combination codes were attached (*de Vries & Dubois, 1983*; *Gudin, 2003*). Hybridizations were formed both in the apical bud and in the flowers formed on the shoots of the axillary buds.

After the last date of hybridization (140 days), fruits that reached harvest maturity, defined as when the color of the fruit changes from green to orange-red, and browning begins on the flower stalk, were harvested. The number of fruits was counted, and the crossability rate (%, CR) per combination (fruit set rate) was determined. The average fresh weight per fruit (FW) for each combination was also recorded using an Ohaus NV 212 model precision weighing machine. Then, the fruits were brought under laboratory conditions, slit with a sharp knife, and the seeds were separated from the fruit to determine the average number of seeds per fruit for each combination (SNpF). The average fresh weight per seed was also recorded (SW) for each combination. Moreover, the data on the number of flowers crossed, fruit and seed sets that make up the fertility indexes were used to calculate the average crossability rate for a parent (ACR), the percentage high crossability for a parent (PHC) and seed production efficiency (SPE).

The ACR for a parent was calculated as the sum of cross compatibility rates in specific crosses divided by the number of cross-combinations involving that particular parent

(*Mondo et al., 2022*):

$$\text{ACR}(\%) = \frac{\sum \text{Crossability rates}}{\text{Number of cross combinations}} \times 100.$$

The PHC for a parent was calculated as the number of times the cross compatibility rate exceeded the species' overall cross compatibility (number of crossability rates > overall mean), $(x)$ divided by the number of cross combinations in which that parental genotype was involved (*Mondo et al., 2022*):

$$\text{PHC}(\%) = \frac{x}{\text{Number of cross combinations}} \times 100.$$

The SPE for a cross was calculated as the number of viable seeds divided by the number of stigmas of the female parent in that cross combination (the expected number of seeds in rose fruits is equal to the stigma number) and the number of pollinated flowers multiplied by 100 (*Mondo et al., 2022*):

$$\text{SPE}(\%) = \frac{\text{Number of viable seed set}}{\text{Number of flower spollinated} * \text{stigma number}} \times 100.$$

The selected $F_1$ hybrid seeds from the fruit were treated with moist cold stratification at $4 \pm 1\,°C$ for 100 days immediately after seed weights were recorded (*Gudin et al., 1990*; *Debener & Mattiesch, 1996*). Perlite was used as stratification medium, and seeds were treated using fungicide with 25% tebuconazole as the active ingredients against fungal diseases for thirty minutes. The seeds were then placed in zip-top bags containing perlite and placed in cold storage. After the moist cold stratification, seeds were sown in vials containing peat and germinated in a plastic-covered greenhouse (at a temperature from 18 to 21 °C) and were irrigated by the fogging irrigation method during the germination process. When the seeds showed cotyledon and hypocotyl growth above the growing medium, they were considered to have germinated (*Nadeem et al., 2015*; *Khan et al., 2020*). The germinated seeds were counted, and the seed germination rate (%, SGR) was determined using the following formula:

$$\text{SGR}(\%) = \frac{\text{Number of seeds germinated}}{\text{Number of seed sown}} \times 100.$$

## The evaluation of pollen quality

The pollen viability and germination rate of all rose species/varieties was determined. Anthers taken from flowers of all rose species and varieties during emasculation were placed in glass bottles and brought to the laboratory. The bottles were kept overnight in a growth chamber in the laboratory for anther dehiscence (in darkness at a temperature of 24 °C and 60% humidity). The pollen's viability and germination rate were determined using dispersed pollen from anthers as soon as the pollen was removed from the chamber. The IKI (iodine potassium iodide) method was used to determine the pollen viability rates, and the petri dish method was used to determine the germination rates (PG). The IKI was applied according to *Doğan et al. (2020)*. Five minutes after being treated with the IKI solution, pollen grains were counted under a microscope. Pollen grains dyed black and dark brown were considered viable, pollen grains dyed orange, red, or light brown

were considered semi-viable, and pollen grains dyed yellow or colorless were considered non-viable. Fifty percent of the pollen grains categorized as semi-viable were accepted as viable. The petri dish method was modified according to *Kazaz et al. (2020)*. A 2-mm layer of germination medium containing 20% sucrose, 10 ppm boric acid, and 1% agar solution was poured into plastic petri dishes. Before freezing, the solution in the petri dishes was divided into four separate areas, and pollen was sprinkled lightly on each area with the help of a brush. The preparation, which was incubated for 8 h at a temperature of 24 °C and 60% humidity, was removed from petri dishes and germinated pollen grains were counted under the microscope. Pollen grains were considered germinated when they formed a pollen tube longer than their own diameter. In both methods, the Leica DM1000 model microscope and imaging system with x40 and x100 magnification objective lenses were used for pollen count.

### Data analysis

The experiments on crossbreeding and pollen quality were established in a completely random design with three replications (*Wasonga et al., 2020*). In the crossbreeding, ten crosses were made in each replication. In the pollen viability test, two coverslips were used for each rose variety, and counts were performed in four areas on each coverslip. In the pollen germination test, counting was performed in four areas over two slices chosen randomly in each petri dish. In both methods, averages of 250 pollen grains were counted per area. Statistical analysis was performed using IBM SPSS Statistics version 20.0 software. An analysis of variance was applied to the angularly transformed data. The mean differences were established using Duncan's test (where $p \leq 0.05$) (*Kılıç et al., 2020*).

The original data on fertility indexes of $F_1$ hybrids were recorded and processed using Microsoft Office Excel 2021 and XLSTAT. To provide a comprehensive evaluation of fertility, the method specified by *Wang et al. (2022)* was used. The fertility indexes for genotypes used as parents and cross combinations were normalized through variable transformation (min-max scaling), and their weight coefficients were calculated using AHP (the consistency rates of parents and combinations were 0.0052 and 0.0004, respectively) and then composite index was calculated. The composite index (CI) is computed by summing the product of each component's value (Xi) and its corresponding weight coefficient (Wi), for all i from 1 to n, where n represents the total number of components. Moreover, a heat map was created to visualize hierarchical clustering and standardized the values for each combination. Principal component analysis was also performed, and a biplot was established for greater approximation than the coefficient of correlation using XLSTAT to determine relationships among the fertility indexes (*Evgenidis, Traka-Mavronaand & Koutsika-Sotiriou, 2011*).

## RESULTS

### Pollen viability and germination rate

The analysis of variance showed that there was a statistically significant difference between pollen parents for pollen germination and pollen viability rates ($p \leq 0.05$). First Red (51.97%) showed the highest pollen viability rate, whereas the lowest viable pollen rate was

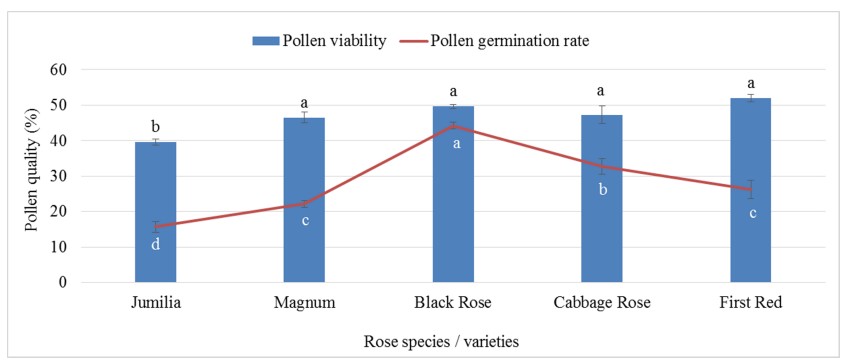

**Figure 1 Pollen viability and germination rates of rose species/varieties (Bars show standard error, $p \leq 0.05$).**

recorded for Jumilia (39.56%). The difference between Black Rose and Magnum, Cabbage Rose, and First Red was found to be statistically insignificant. Similar to viable pollen rates, the highest pollen germination rate was found in Black Rose (44.22%), followed by Cabbage Rose (32.70%), whereas the lowest germination rate was recorded for Jumilia (15.65%) (Fig. 1). First Red had 31.37% more viable pollen than Jumilia. The germination rate of Black Rose pollen grains was 2.83 times higher than that of Jumilia.

## Fertility indexes

The results of the statistical analysis showed that the crossability rate, the average number of seeds per fruit for each combination, the average fruit and seed weight per combination, the seed germination rate, the seed production efficiency, the average crossability rate, and the percentage of high crossability varied significantly among all cross-combinations (Table 3) and species/varieties (Table 4) ($p \leq 0.05$).

In the study, 600 crosses were created, and a fruit set was achieved in 51.16% of them. The highest crossability rate was observed in BR × FR (94.36%), which showed no statistically significant difference with BR × J (90.75%) and CR × J (80.0%). However, the crossability rate was above 65.0% in all combinations where BR was the female parent. The lowest crossability rate was recorded in J × FR (8.70%), and there was no statistical difference between it and J × M (11.68%), M × FR (12.52%), and CR × M (23.12%). Moreover, the crossability rate changed from 32.0% to 60.0% in combinations where BR was the pollen parent, 46.10% and 68.81% where CR was the pollen parent, 48.54% and 90.75% where J was the pollen parent, 11.68% and 76.95% where M was the pollen parent, and 8.70% and 94.36% where FR was the pollen parent (Table 3). The crossability rate of BR × FR was 10.9 times higher than J × FR combinations.

The maximum seed number per fruit was determined to be in FR × CR (19.31 pcs) and was not statistically different from M × BR (16.51 pcs) and M × CR (17.45 pcs). The lowest SNpF were found in J × M (2.12 pcs), J × FR (3.46 pcs), BR × M (4.53 pcs), CR × BR (5.31 pcs) and BR × CR (5.84 pcs) combinations, respectively, and they were not significantly different. The SNpF varied between 5.31 and 16.51 in combinations where BR was the pollen parent, 5.84 and 19.31 where CR was the pollen parent, 6.42 and 12.40

**Table 3  Fertility indexes of cross combinations.**

| Cross-combinations | Cr (%) | SNpF (pcs) | FW (g) | SW (mg) | SGR (%) | SPE (%) |
|---|---|---|---|---|---|---|
| J × M | 11.68 i | 2.12 ı | 2.77 h | 33.48 c | 47.73 a | 0.18ı |
| J × BR | 56.72 d-f | 9.56 b-f | 14.60 a | 64.20 a-c | 8.78 g | 4.03 de |
| J × CR | 55.89 d-f | 9.12 b-f | 7.62 c-e | 38.33 c | 30.69 d | 3.78 d–f |
| J × FR | 8.70 i | 3.46 hı | 3.65 f-h | 48.44 bc | 20.98 e | 0.22ı |
| M × J | 56.19 d-f | 12.40 b | 5.98 d-g | 51.72 bc | 31.71 cd | 4.28 de |
| M × BR | 32.00 hı | 16.51 a | 5.45 e-h | 47.40 bc | 36.84 b-d | 3.28 d-g |
| M × CR | 50.00 e-g | 17.45 a | 8.36 bd | 88.06 a | 31.66 cd | 5.25 d |
| M × FR | 12.52 i | 7.19 d-h | 3.92 f-h | 48.86 bc | 36.73 b-d | 0.51 ı |
| FR × J | 48.54 fg | 11.09 b-d | 6.38 d-f | 73.24 ab | 19.70 ef | 1.34 g-ı |
| FR × M | 46.97 f-h | 8.29 b-g | 3.81 e-h | 75.38 ab | 13.42 fg | 0.95 hı |
| FR × BR | 36.61 g-ı | 11.62 bc | 9.33 bc | 61.50 a-c | 33.94 cd | 1.05 hı |
| FR × CR | 46.10 f-h | 19.31 a | 10.29 b | 56.23 a-c | 31.38 cd | 2.23 e-ı |
| BR × J | 90.75 ab | 7.59 c-h | 4.20 f-h | 39.41 c | 39.08 bc | 17.20 b |
| BR × M | 76.95 bc | 4.53 g-ı | 3.78 f-h | 38.98 c | 33.35 cd | 8.67 c |
| BR × CR | 68.81 cd | 5.84 e-ı | 3.56 gh | 57.44 a-c | 36.40 b-d | 10.01 c |
| BR × FR | 94.36 a | 8.54 b-g | 4.35 f-h | 31.66 c | 41.73 ab | 19.46 a |
| CR × J | 80.00 a-c | 6.42 e-h | 2.67 h | 51.19 bc | 9.14 g | 2.86 e-h |
| CR × M | 23.12 ıi | 9.60 b-f | 3.52 f-h | 52.54 bc | 16.61 ef | 1.26 g-ı |
| CR × BR | 60.00 d-f | 5.31 f-ı | 3.20 gh | 74.67 a-c | 12.79 fg | 1.80 f-ı |
| CR × FR | 65.86 c-e | 9.80 b-e | 3.06 gh | 61.46 a-c | 14.21 e-g | 3.69 d-f |

Notes.
$p \leq 0.05$.

J, Jumilia; M, Magnum; BR, Black Rose; CR, Cabbage Rose; FR, First Red; Cr, crossability rate; SNpF, seed number per fruit; FW, average fruit weight; SW, average seed weight; SGR, seed germination rate; SPE, seed production efficiency.

where J was the pollen parent, 2.12 and 9.60 where M was the pollen parent, and 3.41 and 9.80 where FR was the pollen parent (Table 3). The seed number per fruit of FR × CR was 9.2 times higher than J × M combinations.

The heaviest fruit was observed in J × BR (14.60 g), and the lightest fruit was recorded in CR × J (2.67 g). These crosses were in the same statistical group as all combinations where FR and BR were female parents. Average fruit weight ranged from 3.20 g to 14.60 g in combinations where BR was the pollen parent, 3.56 g to 10.29 g where CR was the pollen parent, 2.67 g to 6.38 g where J was the pollen parent, 2.77 g to 3.81 g where M was the pollen parent, and 3.06 g to 4.35 g where FR was the pollen parent (Table 3). The combination with the highest seed weight was M × CR (88.06 mg). It was in the same statistical group as CR × BR (74.67 mg), J × BR (64.20 mg), CR × FR (61.46 mg), BR × CR (57.44 mg), and in all combinations where FR was the female parent. The lowest seed weight was determined in BR × FR (31.66 mg). Average seed weight changed from 47.40 mg to 74.67 mg in combinations where BR was the pollen parent, 38.33 mg to 88.06 mg where CR was the pollen parent, 39.41 mg to 73.24 mg where J was the pollen parent, 38.98 mg to 75.38 mg where M was the pollen parent and 31.66 mg to 61.46 mg where FR was the pollen parent (Table 3). While the M × CR combination provided 64.05% heavier

**Table 4  Fertility indexes of parents.**

| Rose species/varieties | ACR (%) | PHC (%) | SPE (%) |
|---|---|---|---|
| Jumilia | 51.06 ± 1.03 c | 62.5 ± 1.32 a | 6.15 ± 0.84 c |
| Magnum | 38.67 ± 1.00 e | 50 ± 1.32 b | 9.99 ± 0.99 b |
| Black Rose | 64.53 ± 0.85 a | 50 ± 2.64 b | 41.51 ± 0.93 a |
| Cabbage Rose | 56.22 ± 1.37 b | 50 ± 1.00 b | 7.20 ± 0.95 c |
| First Red | 44.96 ± 0.80 d | 62.5 ± 1.00 a | 4.17 ± 0.92 d |

Notes.
$p \leq 0.05$.
ACR, average crossability rate; PHC, percentage high crossability; SPE, seed production efficiency.

seeds than the BR × FR combination, 81.72% lighter fruit was obtained from the CR × J combination compared to the J × BR combination.

The maximum seed germination rates were observed in J × M (47.73%) and BR × FR (41.73%). The minimum seed germination rate was recorded in J × BR (8.78%), which was not statistically different from CR × J (9.14%), CR × BR (12.79%), FR × M (13.42%), and CR × FR (14.21%). The seed germination rates varied from 8.78% to 36.84% in combinations where BR was the pollen parent, 30.69% to 36.40% where CR was the pollen parent, 9.14% to 39.08% where J was the pollen parent, 13.42% to 47.73% where M was the pollen parent, and 14.21% to 41.73% where FR was the pollen parent (Table 3). Seeds from J × M combinations germinated at a rate 5.44 times higher than seeds from J × BR combinations.

The highest seed production efficiency was found in BR × FR (19.46%), followed by BR × J (17.20%), whereas the lowest seed production efficiency was seen in J × M (0.18%), which was in the same statistical group as J × FR (0.22%) and M × FR (0.51%) and all combinations where FR was the female parent. The seed production efficiency ranged from 1.05% to 4.03% in combinations where BR was the pollen parent, 2.23% to 10.01% where CR was the pollen parent, 1.34% to 17.20% where J was the pollen parent, 0.18% to 8.67% where M was the pollen parent, and 0.22% to 19.46% where FR was the pollen parent (Table 3). The seed production efficiency of BR × FR combinations was 108.11 times higher than that of J × M combinations.

In combinations in which BR was the female parent, 1.45 to 2.49 times more fruit set was achieved, 1.09 to 2.85 times more germinated seeds were obtained, and greater seed production efficiency was found, 4.16 to 9.96 times compared to the combinations in which other genotypes were female parents. In general, the fruit set rate, seed set rate, and seed production efficiency were found to be approximately 2 times, 1.5 times, 1.5 times, and 5 times higher in combinations where old garden roses were used as parents than in cross combinations among modern roses. In combinations where old garden roses were used as pollen parents, the seed set rate was found to be 1.5 times higher than in combinations in which modern roses were used as pollen parents, while the crossability rate and seed production efficiency decreased by 1.2 times. In combinations where old garden roses were used as female parents, the crossability rate and seed production efficiency were found to

be 2 times and 3.6 times higher, respectively, than in combinations in which modern roses were used as female parents.

Among the parents, the maximum ACR rate of 64.53% and the maximum SPE rate of 41.51% were obtained from the Black Rose. The Magnum had the lowest ACR rate of 38.67%, while First Red had the lowest SPE rate of 4.17%. Jumilia and First Red showed a higher PHC rate than the other three parents (Table 4).

## Principal component analysis, correlation matrix, and hierarchical clustering heat map

Principal component analysis (PCA) was carried out and a biplot was established for a better approximation than the coefficient of correlation to describe the crossability success between and/or within rose species and varieties. Given an eigenvalue larger than 1, the first (F1), second (F2), and third (F3) principal components accounted for 38.28%, 20.88%, and 16.28% of the total variation, totaling 75.44%. The PCA-biplot results indicated a positive correlation between pollen germination, fruit weight, seed weight, and seed number per fruit because they were placed on the same side and had similar vector lengths. Higher crossability rates, seed production efficiency, and seed germination rates were found in all cross-combinations where the Black Rose was the female parent compared to other combinations. The stigma number and seed weight were significant parameters in all combinations, where First Red was the seed plant with high values of these traits (Fig. 2).

The correlation matrix supported the relationships obtained by PCA. When the correlation matrix is examined in Table 5, it is seen that there is a positive correlation between crossability rate and seed production efficiency, between seed weight and stigma number, between fruit weight and seed number per fruit, between fruit weight and pollen germination rate, and between seed number per fruit and stigma number. Additionally, it was determined that there was a negative relationship between seed weight and seed germination rate and between stigma number and seed production efficiency.

Heat map analysis based on the fertility indexes of different cross combinations divided the examined traits into two main groups. While stigma number, pollen germination rate, seed and fruit weight, and number of seeds per fruit were included in the 1st major group, fruit set rate, seed production efficiency, and seed germination rate parameters were included in the 2nd major group. Cross combinations were also divided into two main groups hierarchically. The 1st major group was located on the left side of the map, where predominantly positive tendencies were represented by green color tones and negative tendencies were represented by red color tones cross combinations in which Black Rose was used as a female parent. They were grouped in a single cluster showing higher values for seed production efficiency, seed germination rate, and crossability rate compared to all other cross combinations. The 2nd major group was divided into two subgroups. It was seen that the combinations in the 1st subgroup consisted of combinations in which hybrid tea roses were used as females and old garden roses as pollen parents. Despite the low stigma number, Black Rose provided higher seed production efficiency. Similarly, the group with the higher stigma number had lower seed production efficiency (Fig. 3).

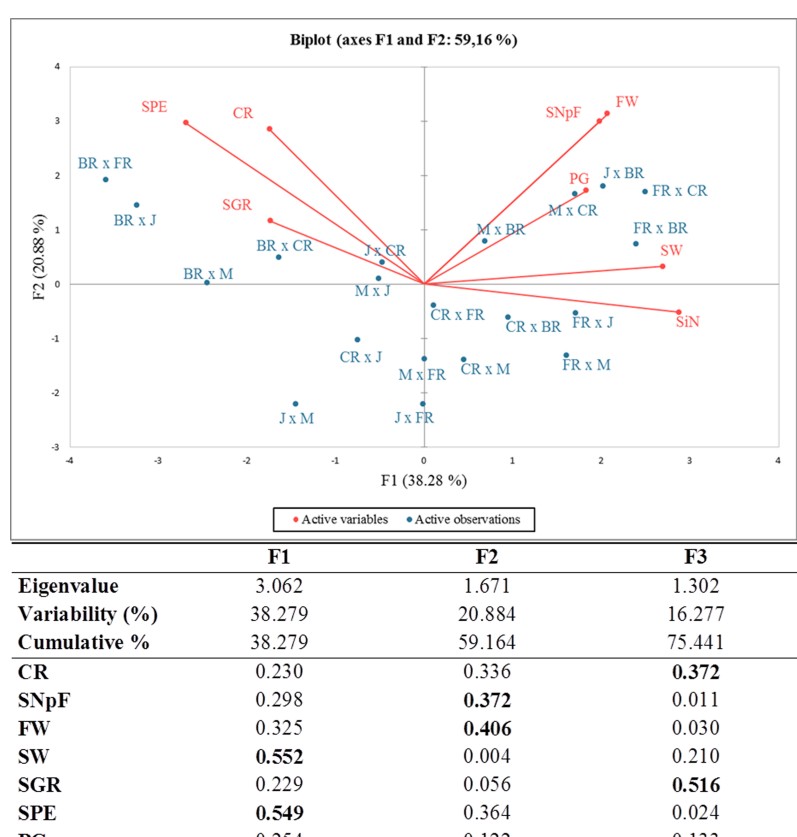

| | F1 | F2 | F3 |
|---|---|---|---|
| Eigenvalue | 3.062 | 1.671 | 1.302 |
| Variability (%) | 38.279 | 20.884 | 16.277 |
| Cumulative % | 38.279 | 59.164 | 75.441 |
| CR | 0.230 | 0.336 | **0.372** |
| SNpF | 0.298 | **0.372** | 0.011 |
| FW | 0.325 | **0.406** | 0.030 |
| SW | **0.552** | 0.004 | 0.210 |
| SGR | 0.229 | 0.056 | **0.516** |
| SPE | **0.549** | 0.364 | 0.024 |
| PG | 0.254 | 0.122 | 0.133 |
| SiN | **0.627** | 0.011 | 0.006 |

**Figure 2** Eigenvalues for principal components, squared cosines of the variables (values in bold correspond for each variable to the factor for which the squared cosine is the largest) and PCA-biplot of cross combinations, fertility indexes. J, Jumilia; M, Magnum; BR, Black rose; CR, Cabbage rose; FR, First Red; CR (red color), crossability rate; SNpF, seed number per fruit; FW, fruit weight; SW, seed weight; SGR, seed germination rate; SPE, seed production efficiency; PG, pollen germination rate; SiN: stigma number.

**Table 5 Correlation matrix (Pearson).**

| Variables | CR | SNpF | FW | SW | SGR | SPE | PG | SiN |
|---|---|---|---|---|---|---|---|---|
| CR | 1 | −0.003 | −0.020 | −0.080 | −0.059 | **0.770** | −0.221 | −0.334 |
| SNpF | −0.003 | 1 | **0.575** | 0.385 | 0.048 | −0.064 | 0.266 | **0.445** |
| FW | −0.020 | **0.575** | 1 | 0.290 | −0.096 | −0.098 | **0.528** | 0.318 |
| SW | −0.080 | 0.385 | 0.290 | 1 | **−0.552** | −0.395 | 0.274 | **0.535** |
| SGR | −0.059 | 0.048 | −0.096 | **−0.552** | 1 | 0.411 | −0.057 | −0.308 |
| SPE | **0.770** | −0.064 | −0.098 | −0.395 | 0.411 | 1 | −0.211 | **−0.594** |
| PG | −0.221 | 0.266 | **0.528** | 0.274 | −0.057 | −0.211 | 1 | 0.127 |
| SiN | −0.334 | **0.445** | 0.318 | **0.535** | −0.308 | **−0.594** | 0.127 | 1 |

Notes.
CR, crossability rate; SNpF, seed number per fruit; FW, average fruit weight; SW, average seed weight; SGR, seed germination rate; SPE, seed production efficiency; PG, pollen germination rate; SiN, stigma number.
Values in bold are different from with a significance level alpha = 0.05.

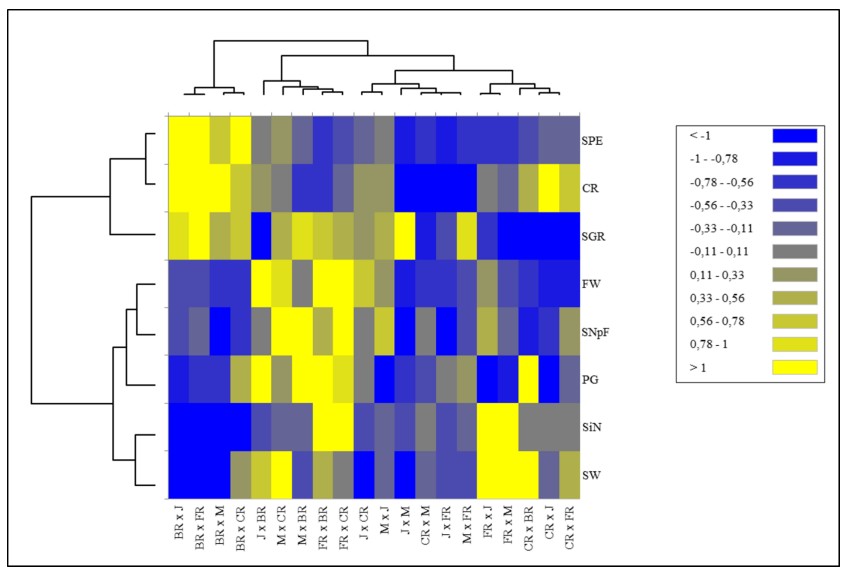

**Figure 3** **Hierarchical clustering heat map of the examined parameters for each of the 20 cross combinations.** A hierarchical clustering heat map was performed on log-transformed quantitative data. The color-coded scale indicates an increase from blue to yellow; J, Jumilia; M, Magnum; BR, Black rose; CR, Cabbage rose; FR, First Red; CR, crossability rate; SNpF, seed number per fruit; FW, fruit weight, SW, seed weight; SGR, seed germination rate; SPE, seed production efficiency; PG, pollen germination rate; SiN, stigma number.

## A comprehensive evaluation of the fertility of the cross combinations and parents

The composite fertility index of parents and cross combinations are shown in Fig. 4. The comprehensive fertility index values of combinations varied from 0.14 to 0.89, making it easier to identify combinations with greater fertility. While the highest comprehensive fertility index was found in 'Black rose × First Red' and 'Black Rose × Jumilia', the lowest fertility index was determined in the 'Jumilia × First Red'. The comprehensive index values of female parents varied from 0.26 to 0.82. While the highest comprehensive fertility index was found in Black Rose as the female parent, the lowest fertility index was determined in Jumilia and Cabbage rose. Magnum and First Red had better comprehensive fertility indexes than Jumilia. The comprehensive index values of the parents as pollen parents varied from 0.13 to 0.90. While the highest comprehensive fertility index was found in Cabbage rose as the pollen parent, the lowest fertility index was determined in Magnum. After the Cabbage rose, Black rose became the second pollen parent with the highest fertility index. Magnum and First Red had better comprehensive fertility indexes than Jumilia.

## DISCUSSION

### Pollen viability and germination rate

Pollen productivity and pollen quality of pollen parents are included in crossbreeding because they are essential in terms of fertilization success and hybridization efficiency. Pollen viability and germination rate parameters, which express pollen quality, must

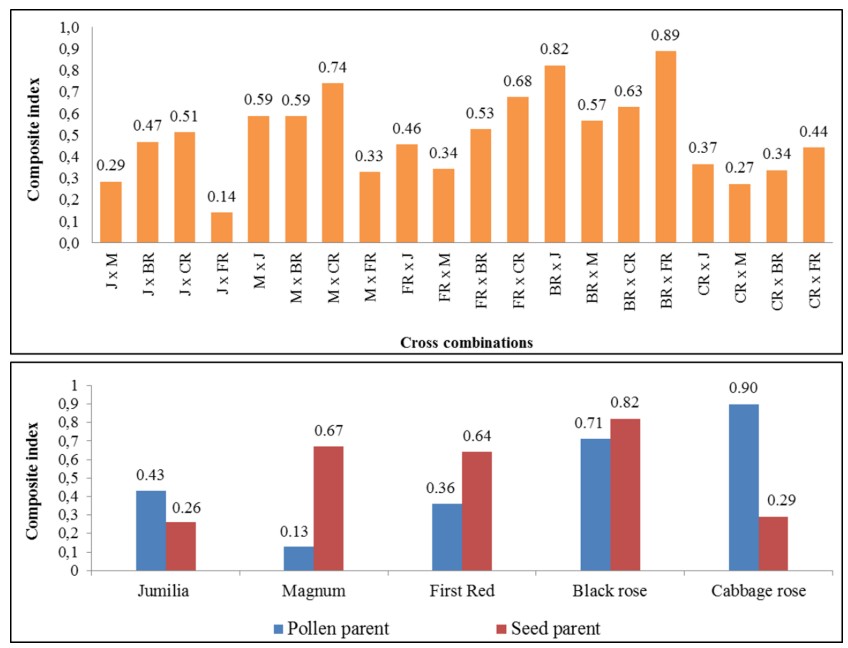

**Figure 4  The comprehensive fertility index of the cross combinations and parents.** J, Jumilia; M, Magnum; FR, First Red; BR, Black Rose; CR, Cabbage Rose.

be high in rose species and varieties for successful fertilization. Various studies have been conducted by many researchers to determine the pollen quality of rose species and varieties. *Pipino et al. (2011)* reported that the germination rates of 11 different hybrid tea roses were between 0% and 46.5%. *Nadeem et al. (2013)* showed that the viability rates of 13 different hybrid tea roses were between 35.0% and 70.0%, and the germination rates were between 1.3% and 46.5%. *Erbaş, Alagöz & Baydar (2015)* stated that in *R. damascena* Mill., pollen viability rates in different flowering periods were between 32.8% and 71.5%, and germination rates were between 24.2% and 57.0%. According to *Żuraw et al. (2015)*, the pollen viability rates of four different rose species ranged from 26.7% to 96.9%. *Giovannini et al. (2017)* reported that the germination rates of 44 different hybrid tea roses varied between 6.0% and 99.0%. *Khan et al. (2021)* stated that the viable pollen rates of some Rosa × hybrida varieties were between 28.60% and 67.40%, and the germination rates were between 6.90% and 67.40%.

In this study, the pollen viability rate varied between 39.56% and 51.97%, while the pollen germination rate varied between 15.65% and 44.22%. Although the findings of this study were generally similar to the results obtained in the studies mentioned, the lower and upper limit values changed among the studies. It is thought that pollen quality varies depending on the genotype, ploidy levels (*Ueckert, 2014*), methods used (*Sulusoglu & Cavusoglu, 2014*), climatic conditions, nutritional status of the plant, time of the pollen collection (season, flowering period, and development period of flowers) (*Martins et al., 2017*), and storage conditions and storage duration (*Miler & Wozny, 2021*). Moreover, similar to the current study results, it has been stated that wild and old garden roses have

higher pollen quality than hybrid roses (*Ueda & Hirata, 1989*; *Gudin & Arene, 1991*; *Zlesak, Zuzek & Hokanson, 2007*; *Kazaz et al., 2020*). The low pollen fertility of hybrid roses could be related to interspecific hybridizations, meiotic abnormalities, heterozygous polyploidy parents, and the accumulation of lethal recessive alleles (*Nadeem et al., 2013*). One of the reasons why old garden roses are so fertile is their ability to produce more morphologically normal pollen. The fact that their germination ability is better than that of all modern roses may be related to the fact that old garden roses are more resistant to dehydration than modern roses. It has been reported that the ability to form morphologically normal pollen in roses and the resistance to dehydration of pollen may vary depending on the species and varieties (*Pacini & Dolferus, 2019*). Another reason is that their capacity to produce 2n pollen may be higher than that of modern roses. *Gao et al. (2019)* indicated that 2n pollen is involved in hybridization and has a competitive advantage while it traverses the stigma and enters the style, and 2n pollen production varies according to genotype in roses.

The pollen quality of the same species/varieties differed because of the chemical and biological test methods, and the viability rates obtained by the IKI method were found to be higher than the germination rates obtained by the agar method in petri dishes. The pollen viability and pollen germination rate ranges varied from each other. In a study by *Parfitt & Ganeshan (1989)*, it was determined that chemical methods did not show similarities with biological methods. Generally, it is expected that there will be a linear relationship between pollen viability and germination rate (*Martins et al., 2017*). However, higher viability rates can be seen compared to the results obtained by biological methods because pollen that has not yet matured can be dyed using chemical methods (*Şensoy et al., 2003*). On the other hand, pollen may retain the capacity to metabolize while losing the ability to germinate (*Gaudet et al., 2020*).

In this study, it was also determined that First Red and Magnum were better than Jumilia in terms of pollen germination rates. This may be because First Red and Magnum are much older than Jumilia. This may be related to the pollen quality of modern roses in varieties developed recently being lower than in older varieties due to the inability to overcome the narrow gene pool problem. Some researchers have stated that hybrid roses are incompatible due to inbreeding. Breeders in the past have used a much smaller number of fertile varieties that produce above-average offspring (*de Vries & Dubois, 1988*).

## Fertility indexes

One of the most important factors affecting success in rose breeding is the fertility of the female parent. While high viability and pollen germination rates are desired in the pollen parents, high fertilization rates, fruit set, seed formation, seed production efficiency, and seed germination rates are expected in the female parents. However, there was variation among the combinations in terms of fruit set, seed formation, seed production efficiency, seed and fruit weight, and seed germination rate.

In this current study, the crossability rate varied from 8.70% to 94.36% among the combinations, and the average crossability rate was 51.16%. The number of seeds per fruit changed from 2.12 to 19.31. In previous research, it was reported that the rate of fruit set in modern roses is generally less than 50% (*Gudin, 2003*), and because of hybridization

between old garden and modern roses, the rate of fruit set decreased to 25% (*Gudin, 2000*). However, *Khan et al. (2021)* determined that the fruit set rate was 63.33% in their crossbreeding study, regardless of the combination. *Abdolmohammadi et al. (2014)* stated that the fruit set rate varied from 0% to 80.00% among the combinations. In the study by *Atram et al. (2015)*, it varied between 0% and 100.0%, and it changed over a range from 30.00% to 83.00% in the *Nadeem et al. (2015)* studies. In studies where the average number of seeds per fruit was determined, it was stated that roses have an average of 0 to 50.0 seeds per fruit (*Zlesak, 2007*). However, *Pipino et al. (2011)* reported that the average number of seeds per fruit in hybrid tea roses varied from 1.10 to 21.30. *Abdolmohammadi et al. (2014)* discovered that the average number of seeds per fruit varied from 0 to 35.30 due to crosses between modern roses and wild and old garden roses. *Nadeem et al. (2015)* determined that the average number of seeds per fruit ranged from 15.0 to 33.0 in combinations using modern roses as parents. *Farooq et al. (2016)* reported that the average number of seeds per fruit varied from 0 to 17.0 in crosses between 5 different rose species. *Khan et al. (2020)* stated that the seed number per fruit ranged from 0.0 to 15.0 among hybrid roses.

The difference in crossability rate among combinations may be related to the complex genetic structures of genotypes (*Ueckert, 2014*), parental fertility (*Nadeem et al., 2015*), incompatibility (*MacPhail & Kevan, 2009*), meiotic abnormalities, and accumulation of lethal alleles (*Ogilvie et al., 1991*; *Nadeem et al., 2015*). Moreover, hormonal control affecting the embryo and hip formation may cause a low crossability rate (*Cruden & Lyon, 1989*; *Stone, Thompson & Dent-Acosta, 1995*). *Gudin (2000)* reported that embryo development seems to control fruit development in roses. In the current study, a positive relationship was found between seed production efficiency and crossability rate. The low crossability rate could be attributed to cross incompatibility since most tetraploid roses are self-compatible (*Rajapakse et al., 2001*). According to (*Zlesak, 2007*), hybrid roses are mostly self-pollinated, and the success of their crossing is dependent on the ability of the female gametes to admit foreign pollen *Nadeem (2012)*. Among the tetraploid parents used in this study, combinations with a low fruit and seed set despite a high pollen germination rate were found where there were combinations with a high fruit and seed set using the same pollen parent. Another reason for the variations in the crossability rate and seed set may be related to petal numbers. It has been reported that roses with a low number of petals form more fruits than those with a higher number of petals, and the number of fruits and seeds is high in the wild, and old garden roses have low petal numbers due to the increase in fertility (*Baydar, Erbaş & Kazaz, 2016*). Researchers also reported that the number of fruits and seeds decreased due to sterility depending on the decrease in the number of anthers as the petal numbers increased. In this study, the lowest number of petals was found in the Black Rose, which showed the highest fruit set rate and had the highest seed production efficiency, which agrees with the study by *Baydar, Erbaş & Kazaz (2016)*. Other reasons are considered for the lower crossability rate and seed set. *Love et al. (2016)* reported that the diameter of the stigma affects the success of crossbreeding, and the number of seeds may vary according to the amount of pollen on the stigma, although it has been determined that an increase in the amount of pollen on the stigma improves the fruit set rate and the number of seeds (*Falque et al., 1995*) while an excessive amount of pollen on the stigma

can decrease the number of seeds (*Lankinen, Lindström & D'Hertefeldt, 2018*). *Ogilvie et al. (1991)* argued that the pollen tube growth barrier could cause decreased seed set.

Fruit and seed weight are important quality parameters in the evaluation of breeding success, and they differed from each other between studies. In this study, fruit weight varied from 2.67 g to 14.60 g, whereas seed weight varied from 31.66 mg to 88.06 mg. However, *Nadeem et al. (2013)* found that the maximum fruit weight was 4.89 g. *Erçişli & Eşitken (2004)* indicated that the fruit weights of different rose species varied from 3.12 g to 5.20 g. *Farooq et al. (2016)* discovered that fruit weights varied from 0 g to 2.0 g when five different rose species were crossbred. *Khan et al. (2021)* determined that fruit weight ranged from 0 g to 5.63 g among hybrid combinations. (*de Vries et al., 2000*) showed that fruit weight varied from 3.50 g to 14.30 g in hybridizations with roses at different temperatures. *Pipino et al. (2011)* stated that the average seed weight of roses was 66.30 mg. *Doğan (2022)* determined that the seed weight of crosses between miniature roses and different rose species and varieties varied from 40 mg to 166 mg. *Turna (2022)* found that the seed weight of roses changed over a range from 120.0 mg to 260.0 mg. In some studies, it has been determined that there is a positive relationship between fruit weight and the average number of seeds per fruit (*Khan et al., 2021*; *de Vries et al., 2000*). In the current study, no relationship was found between fruit weight and seed number or seed weight. It is considered that fruit weight varies depending on the genetic structure of the female parent, and the fruit weight may be largely related to the thickness of the fruit flesh. Moreover, *Pipino et al. (2013b)* indicated that tetraploid hybrids have similar seed and hip development under the same climatic conditions. Contrary to what was stated by the researchers, fruit and seed development differed under the same conditions in this study. This indicates that the genetic structure of the female parent is more effective than the climatic conditions.

Generally, it has been reported that seed germination rates in roses vary from 30% to 45% (*Leus et al., 2018*). In this study, the germination rate varied from 8.78% to 47.73%. However, *Grossi & Jay (2002)* found that the seed germination rate ranged from 0% to 100% depending on the ploidy levels of the parents, and the average seed germination rate was determined to be 14.21% in 112 different hybrid combinations using modern roses as the female parent and both modern and wild roses as the pollen parent. *Pipino et al. (2011)* determined that the seed germination rates of 11 different hybrid tea roses ranged from 15.4% to 37.1%. It was determined by *Ueckert (2014)* that the seed germination rate ranged from 10.6% to 62% in hybrid combinations created with genotypes with different ploidy levels obtained through intraspecific and interspecific hybridization. *Abdolmohammadi et al. (2014)*, because of hybridization studies between old garden and modern roses, found that the seed germination rate ranged from 0% to 93.40%, and the average seed germination rate was 43.41%, depending on the ploidy levels of the parents. *Uran (2022)* stated that seed germination rates of miniature rose × cut rose combinations ranged from 0.00% to 47.50%, but the average germination rate was found to be 10.73%.

The reasons for the differences in the seed germination rate of combinations could be dormancy (*Alp et al., 2009*), plant physiology and morphology (genotype number of female organs, *etc.*; (*Australian Government 2009*)), in addition to the complex genetic

structures of genotypes, parental fertility, incompatibility, meiotic abnormalities, and accumulation of lethal alleles (*Ogilvie et al., 1991*; *Nadeem et al., 2015*). Genetically sterile hybrid seeds may have formed when haploid or triploid pollen grain production occurs in tetraploid pollinating parents. Alternatively, the fact that the response to many applications (hot and cold stratification temperatures and times, *etc.*) for the elimination of dormancy varies depending on the species and varieties (*Alp et al., 2009*), which may have caused differences in the seed germination rate of the combinations. *Lammerts (1946)* reported that the breeding of roses is occasionally hampered by premature abortion of the developing embryo, resulting in few or no viable seeds. Because roses are highly heterozygous, this behavior reduces the efficiency of breeding programs and genetic understanding (*Gudin & Mouchotte, 1996*).

As seen above, many hybridization studies have been conducted on roses, and it has been stated that important parameters such as fruit set rate, the average number of seeds per fruit, and seed germination rate vary considerably among the hybrid combinations (*Gudin, 2000*; *Nadeem et al., 2013*; *Nadeem et al., 2015*; *Fibrianty & Kurniati, 2019*). There are similarities between previous studies and this study, but there are differences between the lower and upper limits in terms of parameters. The reasons for the differences in the lower and upper limits could be the complex genetic structures of genotypes, ploidy levels (*Ueckert, 2014*), parental fertility (*Nadeem et al., 2015*), incompatibility, meiotic abnormalities, accumulation of lethal alleles, climatic conditions (*MacPhail & Kevan, 2009*), dormancy (*Falque et al., 1995*; *Khan et al., 2020*), stratification methods for seeds, plant physiology and morphology, and pollination methods (*MacPhail & Kevan, 2009*).

In the current study, the average number of seeds per fruit of some combinations obtained under *in vivo* conditions was in parallel with the pollen germination rate obtained under *in vitro* conditions. However, the average number of seeds per fruit and the seed production efficiency of some combinations were reduced despite using a pollen parent that had a high pollen germination rate. *Pipino et al. (2011)* indicated that, in some cases, the *in vitro* germination ability may not fully reflect the *in vivo* germination ability. It is thought that this may be related to meiotic abnormalities or pre-pollination barriers in roses. Tetraploid pollen parents are expected to produce pollen grains with a diploid (2n) genome. However, sometimes, due to the abnormalities that occur during the meiotic division of the pollen mother cell, haploid (n) and triploid (3n) pollen grains may also occur besides diploid pollen grains. Because the callus plates of diploid pollen grains are thinner than those of haploid pollen grains, these pollens are more likely to fertilize the egg by moving more easily and faster in the stigma. However, the pollen tube of haploid pollen grains is shorter than the pollen tube of diploid pollen grains and may not reach the ovary within the time required to fertilize the egg cell (*Gao et al., 2019*). Therefore, the rose genotypes used as pollen parents in the current study may have produced haploid pollen grains, and haploid pollen grains germinating *in vivo* may not have shown the same performance under *in vitro* conditions. If incompatible, the pollen grains cannot germinate on the stigma, or even if they germinate, they cannot develop in the stigma and reach the ovary (*Karaağaç & Kar, 2016*). Therefore, every germinated pollen grain in roses should
not be expected to produce seeds, whether due to meiotic abnormalities, pre-pollination barriers, or incompatibility.

Differences were observed between the parents in terms of ACR, PHC, and SPE. The ACR of old garden roses as parents was higher than that of the hybrid tea roses. Old garden roses and Magnum showed more stable crossability and were closer to average. The SPE was higher in the Black Rose. These results indicate that species or varieties with a high crossability rate as the female parent may not always show high seed production efficiency. However, the crossability rate and seed production efficiency of the combinations had positive correlations with each other according to PCA and correlation matrix. These varying relationships between parents and combinations underline the importance of pollen fertility. The ACR and PHC indices provided an overall parental evaluation without distinguishing between female and pollen parents. However, the comprehensive fertility index provided a more effective evaluation that distinguishes between the female and pollen parents. The comprehensive fertility index of female parents indicated that Black Rose was the best female parent followed by Magnum and First Red, in accordance with the SPE value given in Table 4. Furthermore, the comprehensive fertility index supported the PHC and ACR values. The PHC was mostly affected by the seed parent according to the comprehensive fertility index; this may mean that parents with high fertility have more stable crossability and are closer to average. The lower ACR in Magnum, First Red, and Jumilia had a lower comprehensive index value as a pollen parent, which may indicate that ACR value is more affected by pollen parent.

The correlation matrix, PCA, and hierarchical clustering heat map results indicated that seed weight was negatively correlated with the seed germination rate and positively correlated with the stigma number. The negative relationship between seed weight and seed germination rate may indicate that germination does not occur because of the decrease in water absorption and air diffusion due to the increase in seed coat thickness and the prevention of embryonic expression. *Mohapatra & Rout (2005)* attributed a low germination rate to physical dormancy, such as a hard pericarp, which prevents the embryo expansion. According to *Phat et al. (2015)*, weak embryos, thick seed coats, and larger air spaces cause poor seed germination. According to the correlation matrix, there was also a negative correlation between the stigma number and SPE. This is because the Black Rose, which has a lower stigma number compared to other species and varieties, produces seeds at similar rates. Although First Red had ten times the number of stigmas as the Black Rose, the average number of seeds per fruit of First Red was only twice that of the Black Rose. This shows that there are some important differences between the number of seeds per fruit and seed production efficiency.

As a result, the old garden rose species had a higher pollen germination rate than commercial modern roses. Crossing success improved as pollen fertility increased. However, female parent fertility improved the crossing success as much as pollen fertility. Although the pollen fertility and stigma number were low, these combinations had higher crossability rates and seed production efficiency. The maximum seed production efficiency and crossability rate were determined in combinations where BR was the female parent, despite the lower number of stigmas and low pollen fertility. They also had more stable
crossability rate. The SNpF of combinations where hybrid rose varieties were female parents and old garden roses were pollen parents was higher than other combinations of the same female parent. This appeared to be related to both the stigma number and the greater incompatibility between the modern rose varieties. Moreover, the seed germination rate decreased in combinations that produced heavier seeds. It is predicted that it may be related to the seed coat and/or caused by disruptions in the development of the endosperm and embryo. The fruit set rate, seed production efficiency, and the average number of seeds per fruit in interspecific crossing were higher than those obtained in intraspecific crossing. Simultaneously, pollen quality in hybrid roses was lower than that in old garden roses. This suggests a stronger pre-pollination barrier in intraspecific crosses than in interspecies crosses. The presence of a higher seed germination rate in intraspecific hybridization suggests a stronger post-pollination barrier in interspecific hybridization. The results indicated that the pre-pollination and post-pollination isolation mechanisms of old garden and hybrid roses differ from each other. Although Magnum had a higher pollen germination rate than Jumilia, showing a lower comprehensive fertility index as pollen parent made the incompatibility more visible in roses. The germination rate of seeds obtained from combinations of old garden roses and the germination rate of seeds obtained from combinations of hybrid tea roses did not differ from each other. However, the presence of post-pollination barriers in old garden roses will increase the chance of obtaining more hybrids using the embryo rescue method. In crosses between modern roses, this chance is low because there is more of a pre-pollination barrier. Although it is a fact that the chance of seed set increases with the higher number of stigmas in modern roses, the seed set rate did not decrease in parallel with the number of stigmas in old garden roses despite the lower number of stigmas and even gave close results. The seed production efficiency is related to cross-compatibility rather than the number of stigmas and pollen germination rates.

This is the first study to show the crossing success of the BR. Moreover, there was no study on seed production efficiency in rose breeding studies. It is thought that this parameter is important for evaluating breeding studies.

## CONCLUSIONS

The results showed that cross combinations, including old garden roses, can increase the success of breeding programs. Old garden roses, especially Black Rose, as seed and pollen parents, have improved fruit and seed set rates. As female parents, Magnum and First Red appear to be relatively productive hybrid tea roses. The comprehensive fertility index, hierarchical clustering heat map and PCA showed that BR × FR, BR × J, BR × M, and BR × CR combinations can be used successfully. The results indicate that the crosses between old garden roses and hybrid tea roses can be more successful compared to the crosses between hybrid tea roses. The PCA suggests that SPE is a more accurate parameter than SNpF in revealing combination success in breeding programs. The seed production efficiency should be demonstrated in future breeding studies. The results obtained from the comprehensive fertility index may indicate that Magnum and First Red cultivars can create more successful

combinations as maternal parents and cabbage rose as paternal parents. This study will contribute to breeding programs because cross combinations can be determined using the available data on parental performance. The combinations selected in this way are more likely to be successful than a random choice. Moreover, parental selection and determination of cross combinations expend a lot of cost, labor, and time in crossbreeding programs. Increasing the parent gene pool makes a great contribution and is convenient for breeders. The high-volume hybridization strategy used by better-funded international breeding programs cannot be repeated by many public-sector national breeding programs. To improve the effectiveness of their breeding programs and provide more new varieties to their market, they can quickly adopt the method of creating fewer and more careful selections of cross combinations and parents.

## ACKNOWLEDGEMENTS

I would like to thank Prof. Dr. Soner Kazaz for the infrastructure provided.

### Funding
The author received no funding for this work.

### Competing Interests
The author declares that they have no competing interests.

### Author Contributions
- Tuğba Kılıç conceived and designed the experiments, performed the experiments, analyzed the data, prepared figures and/or tables, authored or reviewed drafts of the article, and approved the final draft.

### Data Availability
The raw measurements are available in the Supplementary File.

### Supplemental Information
Supplemental information for this article can be found online at http://dx.doi.org/10.7717/peerj.15526#supplemental-information.

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
