# Peer review of "Identifying successful combinations by fertility index in old garden roses and hybrid tea roses crosses"

_PeerJ, doi:10.7717/peerj.15526_

## Round 0.1 · original submission · Minor Revisions

Dear Author,

This manuscript was a manuscript you submitted earlier and was rejected. But you made an effort to improve it, and we reconsidered it for handling. I evaluated it as a new submission and sent it to the reviewers. Their comments are attached. Your manuscript will be considered for acceptance if you make appropriate corrections.

Reviewer 1 ·

Basic reporting

This study investigated the success of rose breeding programs, which are often low due to poor seed sets and germination rates. The author found that old garden roses can be used as parents to increase the success of rose breeding programs. The author also found that seed production efficiency (SPE) is a more accurate parameter than seed number per fruit (SNpF) in demonstrating combination success in breeding programs of rose.
The experiment was designed well. Abundant data was collected. English language can be improved, typos can be found. The authors should proofread the manuscript carefully to catch any other errors. Overall, this is a well-conducted study with meaningful findings. My detailed comments can be found in the following.

Experimental design

Line 53: What did these numbers imply? SPE?
Line 70: Do you mean rose is the most popular cut flower?
Line 91: 100 per plant/fruit?
Line 196: There are some problems with the equation. Some characters on the left of ‘crossability rates’ can’t show. It should be multiplication sign instead of the letter ‘x’.
Line 202: You can define ‘x’ as ‘the number of times the cross-compatibility rate exceeded the species’ overall cross compatibility’, then put ‘x’ in the equation instead of ‘Number of crossability rates > overall mean’, which doesn’t make sense.

Validity of the findings

Line 289: What do author imply by ‘in the same statistical group’? Please revise the English language here.
It seems that Tables were not named by the order they appeared in this manuscript, please also check the names of the figures.
Line 359: Author presented a PCA-BIPLOT, which means only F1 and F2 were used to represent the variances. Why did the author sum 3 principal components as 75.44%?
Figure 2, line 6: It seems ‘seed number’ was not presented in this figure. Again, F3 should not be presented since it’s not included in the PCA biplot.
Line 423: instead of ‘the lower and upper limit values changed’, can be revised as ‘the pollen viability rate can be in a wider range’.

·

Basic reporting

The manuscript deals with conventional breeding among old roses and new hybrids. The manuscript does not in Materials and Methods describe what were the species of Rosa were taken for the study?

Plants material used for breeding as well varieties are not enough is not enough and breeding results are difficult to attained.

Title need to be revised

Objective need to be revised

Experimental design

no comment

Validity of the findings

Finding cannot describe the clear successful results keeping in view the given objective of this breeding programme

Additional comments

Spelling and punctuation errors should be modified.
Shape titles should be presented in full and clear, terms and phrases should be
described.

---

## Round 0.2 · accepted · Accept

Dear Author,

There are some pretty minor technical errors, but I think you can fix them in the galley proof stage. I have forwarded this situation to PeerJ Staff. They can ask you for new corrections if necessary.

When writing the formulas, he put the "∗" sign instead of the "x" . Please, add the "x" sign from the add icon button.

I am pleased to inform you that after the last round of revision, the manuscript has been improved a lot, and it can be accepted for publication.

Congratulations on accepting your manuscript, and thank you for your interest in submitting your work to PeerJ.

With Thanks